# An Assessment of Microplastic Contamination in a Commercially Important Marine Fish, *Harpadon nehereus* (Hamilton, 1822)

Kalpana Prusty [1], Vasantkumar Rabari [1], Krupal Patel [2], Daoud Ali [3], Saud Alarifi [3], Virendra Kumar Yadav [4], Dipak Kumar Sahoo [5,*], Ashish Patel [4,*] and Jigneshkumar Trivedi [1,*]

1 Animal Taxonomy and Ecology Laboratory, Department of Life Sciences, Hemchandracharya North Gujarat University, Patan 384265, Gujarat, India; kalpanaprusty2021@gmail.com (K.P.); rabarivasant016@gmail.com (V.R.)

2 Marine Biodiversity and Ecology Laboratory, Department of Zoology, Faculty of Science, The Maharaja Sayajirao University of Baroda, Vadodara 390002, Gujarat, India; kjpatel8460@gmail.com

3 Department of Zoology, College of Science, King Saud University, P.O. Box 2455, Riyadh 11451, Saudi Arabia

4 Department of Life Sciences, Hemchandracharya North Gujarat University, Patan 384265, Gujarat, India; yadava94@gmail.com

5 Department of Veterinary Clinical Sciences, College of Veterinary Medicine, Iowa State University, Ames, IA 50011, USA

* Correspondence: dsahoo@iastate.edu (D.K.S.); adpatel@ngu.ac.in (A.P.); jntrivedi@ngu.ac.in (J.T.)

**Abstract:** Microplastic (MP) pollution is a prevalent and global threat to fish. MP contamination was investigated in *Harpadon nehereus* collected from the principal fishing harbors of India's northwest coast. A total of 213 specimens were collected from the major fishing harbors of Gujarat state (Jakhau, Okha, and Jaffrabad) and Maharashtra state (Mumbai). In the laboratory, the morphometric parameters of the specimens, such as total length and body weight, were measured. The collected specimens were analyzed for MP isolation using the previously documented protocol. MPs were quantified under a stereomicroscope, and physical parameters were recorded. All the examined specimens were found to be contaminated with MPs, with an abundance of $6.98 \pm 6.73$ MPs/g. The maximum contamination of MPs was recorded at the study site in Jaffrabad, followed by Jakhau, Mumbai, and Okha. Morphometric analysis of MPs revealed threads to be the most dominant shape. Black and blue MPs with a 1–2 mm size were the predominant recorded types. The chemical composition of extracted MPs revealed polyethylene (PE), polystyrene (PS), and polyurethane (PU) as polymer compositions. Conclusively, the findings highlighted a greater menace to seafood safety due to trophic transfer, which causes a hazardous effect on human health.

**Keywords:** ATR-FTIR; Indo-Pacific region; marine environment; microplastic contamination; seafood safety

**Key Contribution:** The findings of the study have shown MP contamination in a commercially important fish, *Harpadon nehereus*, highlighting a greater menace to seafood safety that causes hazardous effects on human health.

## 1. Introduction

Over the past few decades, plastic manufacturing and usage have drastically expanded due to its low cost, light weight, moisture resistance, tensile strength, and longer durability [1]. According to Plastic Europe (2022), the global annual production of plastic was reported to be 390.7 million metric tons in 2021 [2]. Due to the inappropriate management of plastic trash, it has reached the oceanic environment in numerous ways, including fishing activities, wastewater discharge, atmospheric transport, and tourism [3]. Annually, there is a discharge of about 0.24 million metric tons (MMTs) of plastic garbage into the sea [4].

This plastic debris undergoes physical and chemical alteration, resulting in different sizes, i.e., macroplastics (>2.5 cm), mesoplastics (0.5–2.5 cm), MPs (>1 μm~<5 mm), and nanoplastics (1–1000 nm) [3–5]. MPs can float in surface water due to their reduced size and higher buoyancy, and they are ultimately carried by wind and water currents over further distances in the ocean [1,6]. MP pollution has been identified as a significant ecotoxicological hazard for marine life. Based on the sources, MP particles can be categorized into two classes: primary MPs and secondary MPs [6,7]. Primary MPs are produced intentionally as microbeads for their application in cosmetics, drug carriers, and industrial abrasives [8,9]. While the formation of secondary MPs takes place due to the deterioration and breakage of large-sized plastics under various environmental physical and chemical factors like ultraviolet radiation, photodegradation, and oxidation [5,8]. The different shapes of MPs include pellets, granules, spherical beads, filaments, films, fragments, and foams [7,10].

To date, MP contamination has been recorded in marine fauna, including amphipods [11], coral [12], sea cucumber [13], mussels [14], shrimp [15], crab [16], and fish [17]. Previously, MPs were recognized as pollutants, but recently they have garnered significant attention as a seafood contaminant. Globally, MP ingestion has been recorded in commercially important fish such as *Boops boops* [18], *Chaeturichthys stigmatias* [19], *H. nehereus* [20], *Eleutheronema tridactylum* and *Clarias gariepinus* [21], *Basilichthys australis* [22], *Mugil cephalus* [22], and *Trichiurus lepturus* [23]. Similarly, MP accumulation has been investigated in commercially important fish such as *Rastrilleger kanagurta* and *Epinephalus merra* [24], *Sardinella longiceps* [25], *Coilia dussumieri* [26], *Cyanoglossus macrostomus* [27], *Decapterus russelli* [28], *R. kanagurta* [29], and *Selaroides leptolepis* [30] inhabiting marine areas of India.

The gastrointestinal tract (GT) of *H. nehereus* exhibited a preponderance of small fish and prawns [31]. Various pieces of literature have revealed that several small fish and prawns in India have been contaminated with MPs [32,33]. The transfer of MPs from small fish and prawns to *H. nehereus* can occur through trophic transfer within the food chain. Therefore, it was quite imperative to assess the presence of MP contamination in the GT of *H. nehereus*. It was found that false ingestion of MPs can reduce the fitness of organisms due to starvation, inflammation of the GT, growth inhibition, oxidative stress, and lower fertility [34,35]. The high volume-to-surface ratio of MPs makes them able to absorb hazardous chemicals [36]. Moreover, long-term exposure to these pollutants has been proven to pass into fish tissue [37]. Consequently, accumulated MPs can be transferred to higher taxa due to bioaccumulation [38]. In addition to this, during the consumption of fish contaminated with MPs by humans, there is a possibility of the transfer of these particles from the fish's tissues to human bodies [39].

India's coastline is 8000 km long and includes more than 60 districts spread over nine coastal states, where one-third of the country's population resides [40]. The annual fish landing in India is estimated to be 3.49 million tons (MTs) in 2022, an increase of 14.53% from 2021's landings [41]. Gujarat state covers the longest coastline in India, with 16 coastal districts [5]. The estimated fish landing in Gujarat state was 5.03 lac tons in 2022 [41], while the neighboring state of Maharashtra has a 720 km long coastline and an annual fish landing of about 1.70 lac tons in 2022 [41]. *H. nehereus*, commonly known as Bombay duck, belongs to the family Scopelidae and is the only species with a significant fishery in the states of Gujarat and Maharashtra in India. The annual landing of *H. nehereus* in India was reported to be 55,342 tons in 2022 [41], where the Gujarat and Maharashtra states alone account for 70% of *H. nehereus* landings in India. Recently, MP contamination has been recorded in the marine environments of Gujarat [3,7] and Maharashtra [42], possibly due to improper plastic waste disposal management, fishing activities, and tourism. Despite knowing the facts of MP prevalence and accumulation in the marine environment of Gujarat and Maharashtra, it was quite imperative to assess the MP contamination in a commercially important fish, *H. nehereus,* of Gujarat and Maharashtra states, India.

## 2. Materials and Methods

### 2.1. Study Area and Specimen Collection

The Gujarat and Maharashtra states cover 29% of the Indian coastline, which supports rich biodiversity and diverse habitats, including rocky shores, mud flats, sandy shores, mangroves, coral reefs, and estuaries [35,38]. Fishing activities, industries, urbanization, and tourism are the dominant human activities prevalent in coastal areas of both states. The fish specimens were collected from major fishing harbors on the northwest coast of India: Jakhau, Okha, Jaffrabad, and Mumbai (Figure 1). Jakhau (23°11′14″ N, 68°38′23″ E) is located in the northwestern part of the Gulf of Kachchh, Gujarat state. It is the oldest port in the Kachchh district and provides export facilities [43,44] Okha (22°28′35″ N, 69°04′11″ E) is a well-known pilgrimage site and plays a significant role in fish export. Salt pan industries and fishing activities dominate this area and release effluents into the sea [45]. Jaffrabad (20°86′67″ N, 71°36′67″ E) is a town in the Amreli district of Gujarat state. Local fishing activities, salt pans, and cement factories provide livelihoods to people [46]. Mumbai (19°02′22″ N, 72°49′55″ E) is a well-known metropolitan city in northwest India. The coastal region of Mumbai is dominated by fishing activities, the fertilizer industry, thermal power plants, cargo handling activities, and the oil and pharmaceutical industries [35,40]. A total of 213 fish specimens (study site Okha, n = 50; study site Jakhau, n = 75; study site Jaffrabad, n = 38; and study site Mumbai, n = 50) of *H. nehereus* were collected from December 2022 to January 2023 from the major fishing harbors on the northwest coast of India. The specimens were immediately placed in ice boxes and transported to the laboratory.

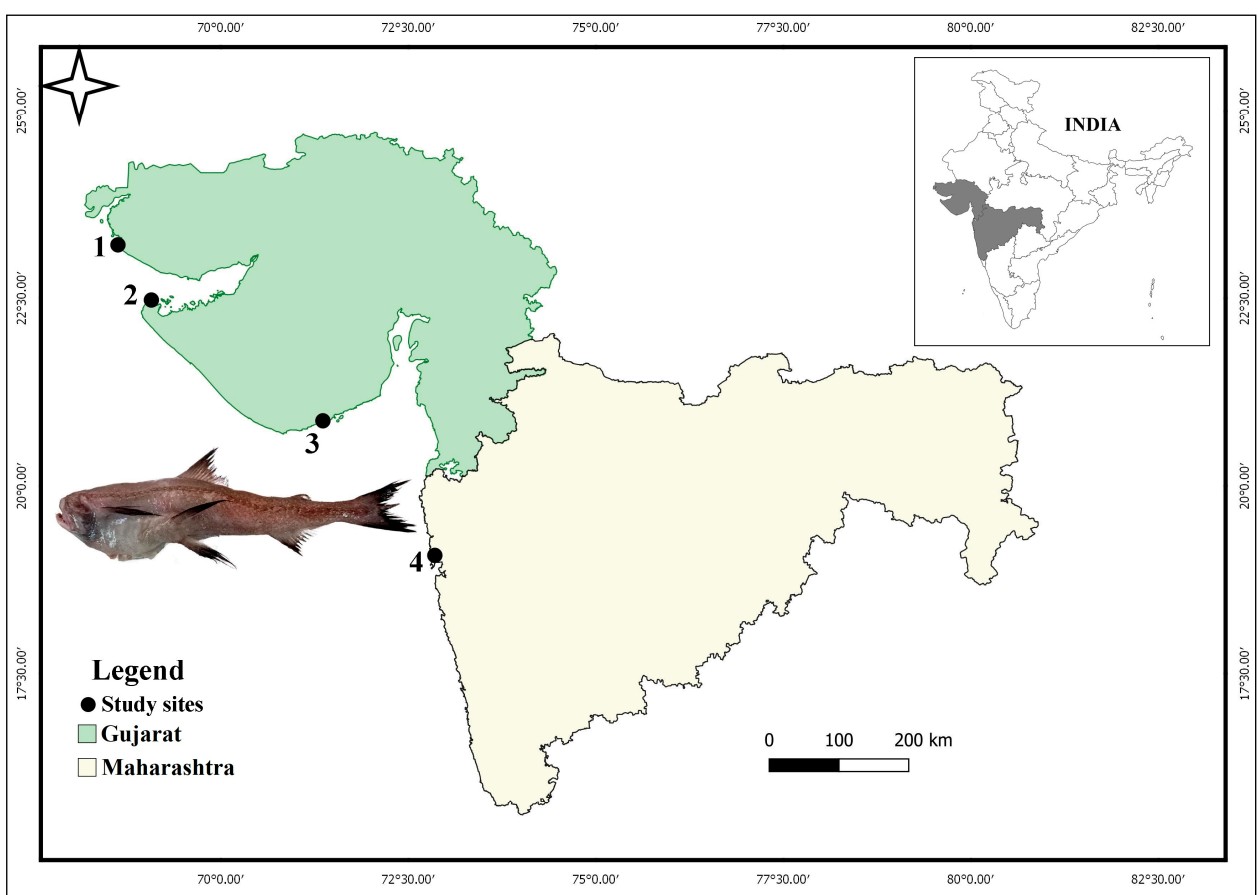

**Figure 1.** Geographic map showing sampling locations: 1—Jakhau, 2—Okha, 3—Jaffrabad, and 4—Mumbai.

## 2.2. MPs Extraction

The specimens were pre-washed with Milli-Q water to eliminate surface adhesive contamination. Morphometric parameters such as the total length (cm) and weight (g) of each fish were recorded. Each fish was dissected in a metal tray, and the GT was isolated to check for contamination with MPs. The weight of the GT of each fish was recorded. The GT of each fish was placed in a beaker, and 10% potassium hydroxide was added to digest organic tissue [25]. Later, it was placed in a hot air oven at 60 °C until complete digestion of organic tissue was achieved [47]. A supersaturated NaCl aqueous solution (1.2 g/mL) was added to float the MPs as per the density gradient [3,5,48], which was followed by the continuous stirring of the solution with a glass rod, after which it was kept at room temperature for 24 h [26]. The supernatant containing floated MPs was filtered through ash-less Whatman filter paper (Grade No. 41, pore size: 20 μm).

## 2.3. Identification and Categorization of MPs

Each filter paper was observed under a stereomicroscope (Metlab PST 901) for identification and quantification of MPs. MPs were visually identified and classified as per their physical characterization, such as shapes (thread, film, foam, and fragment), colors, and size classes (1–2 mm, 2–3 mm, 3–4 mm, and 4–5 mm) [49]. The identification of functional groups and chemicals in the MPs was analyzed using Fourier transform infrared spectroscopy (FTIR). The FTIR measurement was done in the range of 599 to 4000 cm$^{-1}$ at a resolution of 1 nm using the Nicolet Model (USA). The spectra were matched with known plastic libraries (FLOPP and FLOPP-e, n = 762 spectra) [50]. Matches with over 70% spectral similarity were considered MPs. Biofouling, oxidation, and fragmentation of MPs can limit the degree of spectral match.

## 2.4. Contamination Control

The glassware and stainless-steel utensils were prewashed with Milli-Q water in order to eliminate potential contamination. Nitrile gloves and clean cotton lab coats were used during the analysis. To avoid airborne plastic contamination, specimens were covered with aluminum foil. Moreover, blank replicates with the salt solution were used to detect contamination. No contamination was found after visual observation of blank filter papers under a stereomicroscope.

## 2.5. Data Analysis

To understand the MP contamination in *H. nehereus*, an abundance and standard deviation were calculated. A Shapiro-Wilk test was incorporated to check the homogeneity of the data. Since the data were not following a normal distribution, a non-parametric test was performed (Figure S1, see "Supplementary Material", $p < 0.001$). A Kruskal-Wallis test was performed to check the variation of MP contamination between study sites. Moreover, a pairwise Wilcox post-hoc test was incorporated to understand the variation of MP contamination within the group (significant level = 0.05). The percentage composition of MP shape, color, and size was calculated. The statistical analysis was carried out using R Studio (version 4.2.3) and MS Excel.

## 3. Results

### 3.1. Abundance of MPs

In the present study, MP contamination was assessed in a commercially important fish, *H. nehereus*, where a total of 213 fish were examined for MPs, and all 213 specimens had contamination with MPs. A total of 3409 MP particles were obtained from the GT of the fish. The highest number of MPs was recorded at study site Jakhau (n = 1382), followed by Okha (n = 943), Jaffrabad (n = 614), and Mumbai (n = 468). The abundance of MP contamination was recorded as 6.98 ± 6.73 MPs/g. The highest amount of MP contamination was recorded in Jaffrabad (8.85 ± 7.51 MPs/g), followed by Jakhau (8.76 ± 8.20 MPs/g), Mumbai (5.54 ± 5.56 MPs/g), and Okha (4.32 ± 1.83 MPs/g) (Figure 2). The abundance of

MP contamination in *H. nehereus* varied significantly between study sites (H ($\chi^2$) = 77.88, *df* = 212, *p* < 0.01). A pairwise Wilcox post-hoc analysis revealed that the abundance of MP contamination in *H. nehereus* varied significantly between study sites Jaffrabad and Mumbai (*p* < 0.01), between study sites Jaffrabad and Okha (*p* < 0.01), between study sites Jakhau and Mumbai (*p* < 0.001), and between study sites Jakhau and Okha (*p* < 0.001) (Table 1). However, the abundance of MP contamination did not vary significantly between study sites Jaffrabad and Jakhau (*p* = 0.85) and between study sites Mumbai and Okha (*p* = 0.85) (Table 1). A negative correlation was observed between fish length and an abundance of MPs ($R^2$ = 0.05; *df* = 212; *F* = 11.33; *p* < 0.001) (Figure 3).

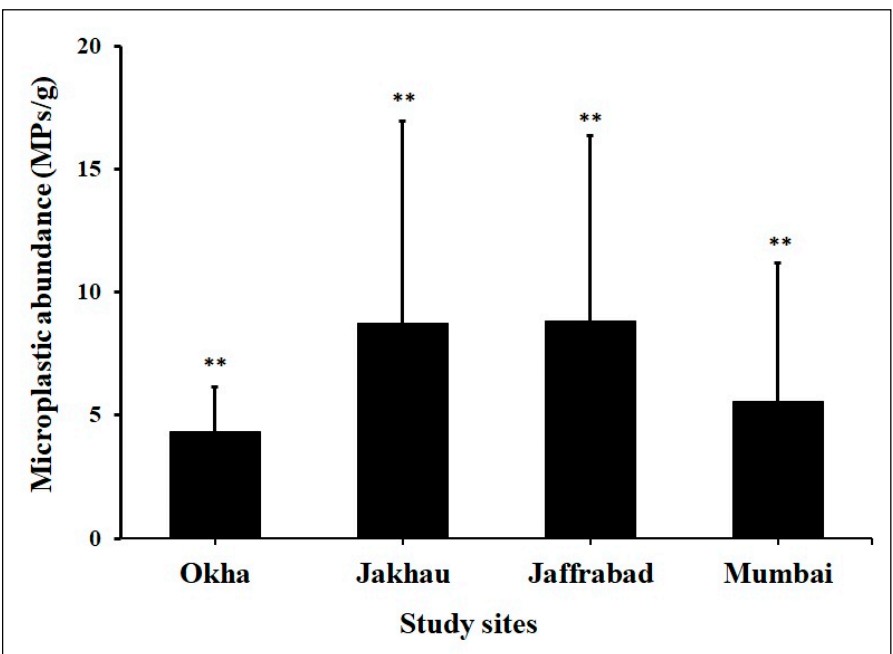

**Figure 2.** An abundance of MP contamination in *H. nehereus* collected from major fishing harbors on the northwest coast of India (** *p* < 0.01).

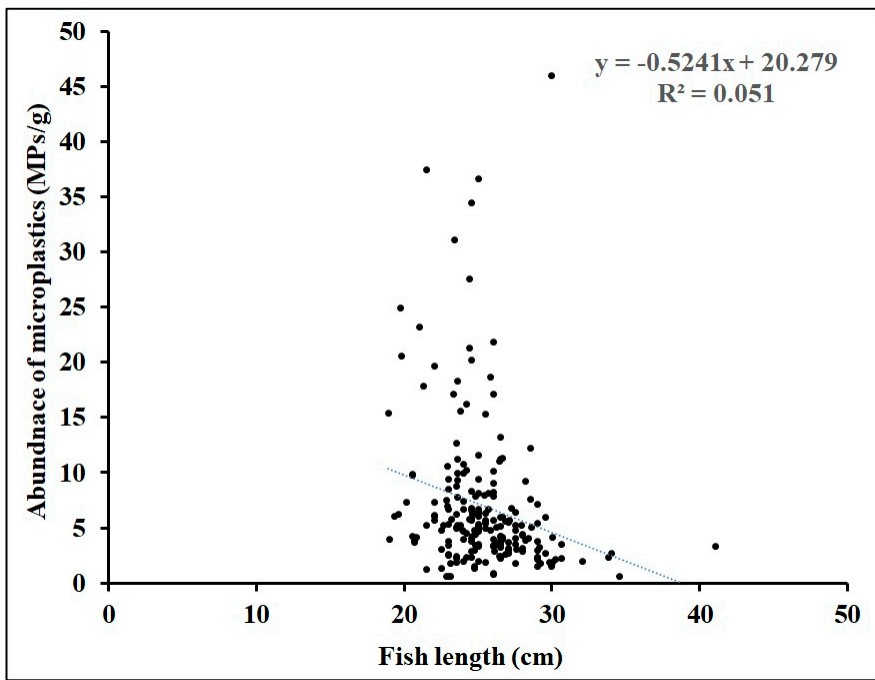

**Figure 3.** Regression analysis between *H. nehereus* body length and MP contamination (*p* < 0.001).

**Table 1.** Results of a post-hoc test showing the variation of MP contamination in *H. nehereus* between study sites.

|  | Jaffrabad | Jakhau | Mumbai |
|---|---|---|---|
| Jakhau | 0.85179 | - | - |
| Mumbai | 0.00172 | 0.00025 | - |
| Okha | 0.00172 | 0.00015 | 0.85179 |

*3.2. Physical and Chemical Characterization of MPs*

The shape classification of MPs revealed the dominance of thread-shaped MPs in all the study sites, followed by fragments, film, and foam (Figure 4). A photograph of each representative shape of MPs was captured under a stereomicroscope (Figure 5). In terms of the color classification of MPs, black, blue, and red MPs were the most prevalent (Figure 6). Size classification of MPs revealed that the 1–2 mm size class was dominant in all the study sites, followed by 2–3 mm, 3–4 mm, and 4–5 mm (Figure 7). The prevalence of MPs in different study sites varied significantly between size classes of MPs (H ($\chi^2$) = 230.01, df = 212, $p < 0.001$). Attenuated total reflectance-Fourier transform infrared spectroscopy (ATR-FTIR) analysis of extracted MPs revealed PE, PS, and PU as polymer compositions of MPs (Figure 8).

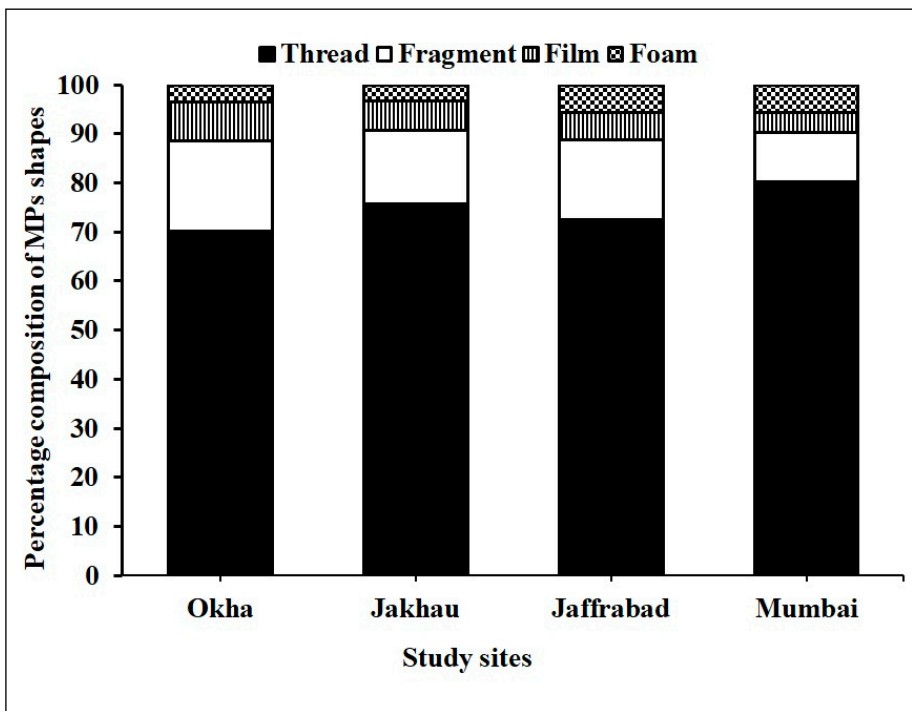

**Figure 4.** Percentage composition of shapes of MPs found in *H. nehereus* collected from major fishing harbors on the northwest coast of India.

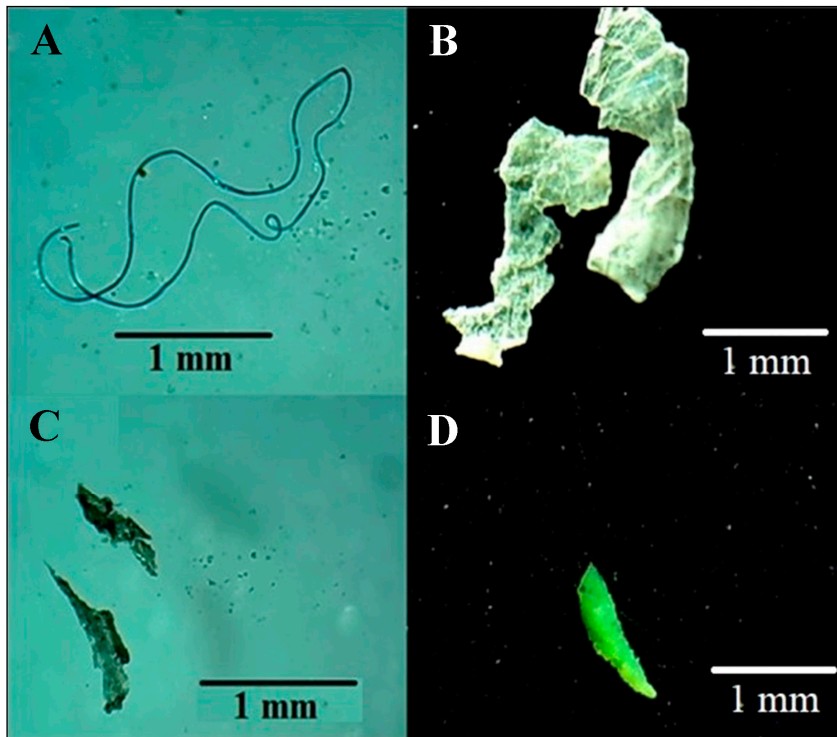

**Figure 5.** Representative photographs of MP shapes found in *H. nehereus* using a stereomicroscope: (**A**) thread; (**B**) foam; (**C**) film; and (**D**) fragment.

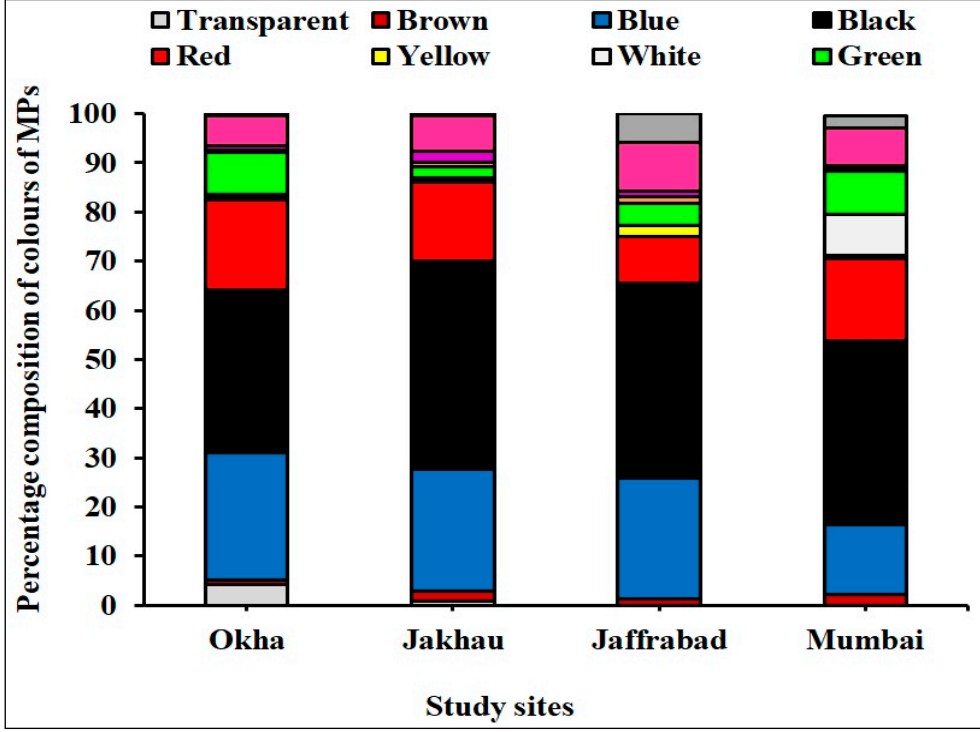

**Figure 6.** Percentage composition of MP colors found in *H. nehereus* collected from major fishing harbors on the northwest coast of India.

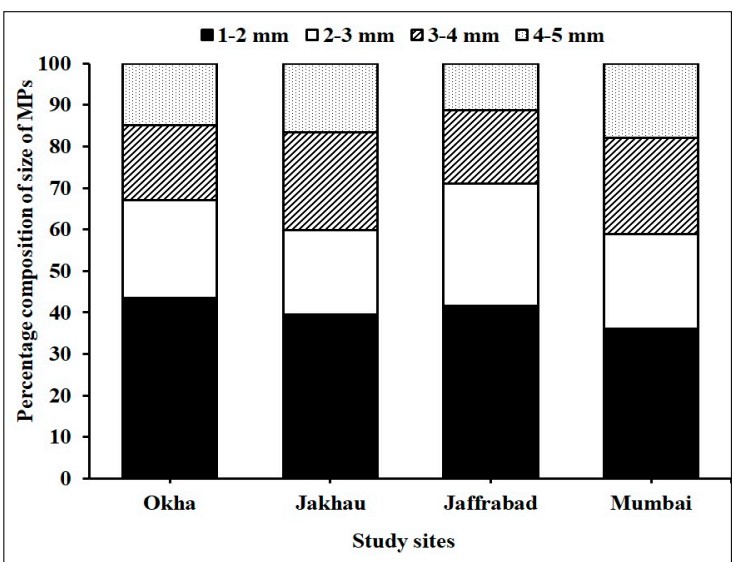

**Figure 7.** Percentage composition of MP sizes found in *H. nehereus* collected from major fishing harbors on the northwest coast of India (*p* < 0.001).

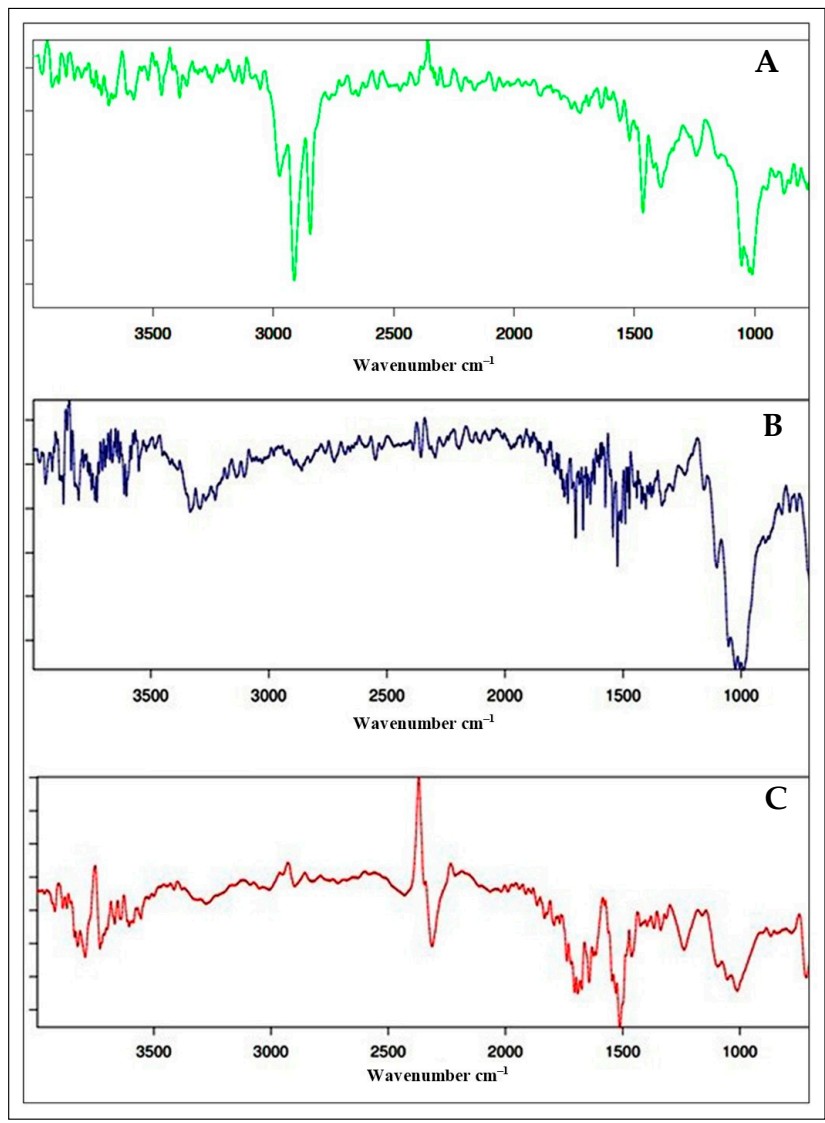

**Figure 8.** ATR-FTIR spectrum of extracted MPs: (**A**) PE; (**B**) PS; and (**C**) PU.

## 4. Discussion

### 4.1. Abundance of MPs in H. nehereus

The present study investigated MP contamination in a commercially important fish, *H. nehereus,* collected from major fishing harbors on the northwest coast of India. The study revealed 100% contamination with MPs in the collected specimens. Similarly, in another study, MP contamination was recorded in all the specimens, including in *Hyporhamphus intermedius, Liza haematocheila, Coilia ectenes, Lateolabrax japonicus,* and *Sillago sihama* collected from Shanghai market, China [51], and *Luciopimelodus pati, Pseudoplatystoma corruscans, Oligosarcus oligolepis, Parapimelodus valenciennes,* and *Odontesthes bonariensis* collected from Rio de la Plata estuary, Argentina [52]. However, less contamination of MPs than what was seen in the present study was recorded in *Solea solea* collected from the Adriatic Sea, Italy (95% of collected specimens) [53], *Engraulis japonicus* collected from Tokyo Bay, Japan (77% of collected specimens) [54], *Argyrosomus regius, Caranx crysos,* and *Dentex dentex* collected from the Mediterranean Sea, Turkey (57.8% of collected specimens) [55], *Etmopterus spinax, Raja clavate, Scyliorhinus canicular,* and *Galeus melastomus* collected from the southern Tyrrhenian Sea, Italy (21% of collected specimens) [56], and *Nemichthys scolopaceus, Arctozenus risso,* and *Xenodermichthys copei* collected from the North Atlantic Ocean (11% of collected specimens) [57].

The abundance of MP contamination in *H. nehereus* was recorded as 6.98 ± 6.73 MPs/g. In another study, the abundance of MP contamination recorded in *T. lepturus* (34.17 ± 3.32 MPs/g) [23], *Lutjanus gibbus* (39.3 ± 4.8 MPs/g), and *M. cephalus* (24.8 ± 7.7 MPs/g) [58] was higher than in the present study. In contrast, less MP contamination was recorded in *A. regius* (0.80 ± 0.8 MPs/g), *Merluccius merluccius* (0.40 ± 0.89 MPs/g), *Scomber japonicus* (0.57 ± 1.04 MPs/g), *Trigla lyra* (0.26 ± 0.57 MPs/g), and *B. boops* (0.09 ± 0.3 MPs/g) [18] than in the present study. The comparison of MP contamination between different species of fish is provided in Table 2. It was found that MP contamination varies between species, possibly due to variations in feeding habits, digestive physiology, position at the trophic level, and size of the organism [49,50]. In the case of intraspecific comparison, the abundance of MPs in *H. nehereus* was higher in the present study than the MP contamination recorded in *H. nehereus* collected from the Northern Bay of Bengal, Bangladesh (0.37 ± 0.10 MPs/g) [20] and Tuticorin, India (0.2 ± 0.03 MPs/g) [30]. According to Bom and Sá F and a team led by Daniel, MP ingestion by the same species can vary with the age of the organism and the geographical location of the study area [33,59].

The possible pathways for MP ingestion by *H. nehereus* are water and food. Filter feeding allows large amounts of water containing planktonic prey to enter the mouth and be released by the gills. As a result of this event, small food particles containing MPs can be retained by gill rakers and eventually transferred to the GT [25,33,60]. Such passive feeding habits by filter feeding make them more susceptible to contamination with MPs, which might be the possible reason for higher MP contamination in filter-feeding fish. In addition to this selective feeding behavior of *H. nehereus*, trophic transfer plays a significant role in MP ingestion. The dietary composition of *H. nehereus* includes fin fish (unicorn cod, sciaenids, gold-spotted anchovy, and juvenile Bombay duck) and shellfish (non-penaeid prawns and penaeid prawns) [31]. Small fish and prawns are more likely to feed on phytoplankton and zooplankton [61], which makes them more prone to ingesting MPs. This event revealed the transportation of MPs in the food web from lower organisms to higher taxa.

**Table 2.** Comparison of MP contamination in different fish species around the world.

| No. | Location | Species | Habitat | Feeding Strategy | MPs in Fish (MPs/g) | Dominant MP Type | References |
|---|---|---|---|---|---|---|---|
| 1 | Northwest coast of India | *H. nehereus* | Pelagic | Carnivore | 6.98 ± 6.73 | 98% Fibers | Present study |
| 2 | Portuguese coast | *A. regius* | Benthopelagic | Carnivore | 0.80 ± 0.8 | 65.8% Fibers | [18] |
| | | *M. merluccius* | Bathydemersal | Carnivore | 0.40 ± 0.89 | | |
| | | *S. japonicus* | Pelagic–neritic | Omnivorous | 0.57 ± 1.04 | | |
| | | *T. lyra* | Benthopelagic | Carnivore | 0.26 ± 0.57 | | |
| | | *B. boops* | Demersal | Omnivorous | 0.09 ± 0.3 | | |
| 3 | Australia and Fiji | *Plectropomus leopardus* | Reef-associated | Carnivore | 45.1 ± 5.1 | 82.4% Fibers | [58] |
| | | *Upeneichthys lineatus* | Demersal | Carnivore | 2.7 ± 0.24 | | |
| | | *L. gibbus* | Benthopelagic | Carnivore | 39.3 ± 4.8 | | |
| | | *M. cephalus* | Benthopelagic | Carnivore, Detrivore | 24.8 ± 7.7 | | |
| 4 | Bangladesh coast Northern Bay of Bengal | *H. translucens* | Demersal | Carnivore, | 1.10 ± 0.30 | 50–55% Fibers | [20] |
| | | *Scorpaenopsis gibbose* | Pelagic | Planktivore | 1.55 ± 0.48 | | |
| | | *H. nehereus* | Pelagic | Carnivore | 0.37 ± 0.10 | | |
| 5 | Haizhou Bay, China | *Thryssa kammalensis* | Pelagic | Carnivore | 11.19 ± 1.28 | 85.7% Fragments | [19,62,63] |
| | | *C. stigmatias* | Pelagic | Predatory | 1.61 ± 0.56 | | |
| 6 | Northeast Atlantic Ocean | *Dicentrarchus labrax* | Demersal | Carnivore | 0.054 ± 0.099 | 54% Fibers | [64] |
| | | *Trachurus trachurus* | Benthopelagic | Carnivore | | | |
| | | *Scomber colias* | Pelagic | Carnivore | | | |
| 7 | Kerala coast of India | *S. longiceps* | Pelagic | Planktivores | 0.054 ± 0.098 | 55.6% Filaments | [25] |
| 8 | Northeast coast, Arabian Sea | *C. dussumieri* | Pelagic | Planktivores | 28.84 ± 10.13 | 54.73% Fibers | [26] |
| 9 | Southeast coast of India | *Istiphorus platypterus* | Epipelagic | Carnivores | 0.0002 ± 0.0001 | 60% Fibers | [30] |
| | | *Sardinella albella* | Reef-associated | Herbivores | 0.75 ± 0.01 | | |
| | | *H. nehereus* | Pelagic | Carnivores | 0.2 ± 0.03 | | |
| | | *Katsuwonus pelamis* | Pelagic | Carnivores | 0.001 ± 0.0005 | | |
| | | *Chirocentrus dorab* | Reef-associated | Carnivores | 0.08 ± 0.04 | | |
| | | *R. kanagurta* | Pelagic | Planktivores | 0.13 ± 0.05 | | |

The study site of Jaffrabad has shown a higher abundance of MP contamination than Jakhau and Okha. Moreover, significant contamination was recorded between the study sites. Similarly, the occurrence of MP particles in marine fish varied significantly between study sites in Pantai Remis and Mersing, Malaysia [65]. The abundance of MP contamination in marine organisms can be influenced by region-specific plastic pollution [51]. The high abundance of MP contamination in the study sites of Jaffrabad and Jakhau is possibly due to extensive fishing activities at both study sites. Moreover, recreational activities and urbanization increase the concentration of plastic trash in the ocean through tidal and wind currents. Regression analysis between fish length and MP abundance revealed a significant negative relationship. In contrast, no relation was recorded between fish length and MP abundance in fish collected from the south coast of India [29,66]. However, apart from this, the level of MP contamination can be influenced by the age and feeding pattern of fish, as well as the level of MP contamination in their living environment.

*4.2. Physical Characterization of MPs*

Regarding the shape classification of MPs, threads were found to be the dominant shape, followed by fragments, film, and foam. Similarly, threads were the dominant shape in *C. dussumieri* [26], *T. lepturus* [23], and *H. translucens* [20]. In contrast, fragments were recorded as the dominant shape in *R. kanagurta* [21], *Chelon subviridis*, *Johnius belangerii*, *R. kanagurta*, and *Stolephorus waitei* [67], *M. cephalus*, *Liza macrolepis*, *S. sihama*, and *Gerres*

*filamentosus* [48], *E. japonicus* [54], and *S. longiceps*, *S. gibbosa*, *Stolephorus indicus*, *R. kanagurta*, and *C. macrostomus* [27]. The prevalence of thread-shaped MPs is closely linked to the dominance of fisheries activities, laundry, textile industries, and wastewater discharge in the study area [55–57].

In the case of the color classification of MPs, black and blue MPs were the dominant colors found in the GT of *H. nehereus*. Similarly, in another study, black and blue MPs were the dominant colors recorded in other fish [33,59,60,64]. It was observed that blue and black trawling nets and ropes are extensively used in fishing activities [29]. The appearance of MPs might trick marine organisms into viewing them as food due to their similar appearance [68]. In the case of size classification of MPs, 1–2 mm-sized MPs were the dominant size, followed by 2–3 mm, 3–4 mm, and 4–5 mm. Similarly, various-sized MPs were found in the GT of fish [24,49,61]. Smaller-sized MPs in the GT of fish may be due to a breakdown of larger debris into smaller-sized particles [66,69]. Moreover, it was found that MPs can act as carriers for other harmful pollutants, such as toxic chemicals (polychlorinated biphenyls, plasticizers, and polybrominated diphenyl ether) and heavy metals, which can attach to the surface of the particles [65,70].

*4.3. Chemical Composition of MPs*

The chemical composition of polymers in the present study was confirmed as PE, PS, and PU. Similarly, PE and PS were recorded as polymer compositions of MPs in commercially important marine fish collected from Malaysia [65]. The chemical composition of extracted MPs is useful for predicting the source of MPs in marine environments [71]. PE is used in toys, cable insulation, grocery bags, housewares, squeeze bottles, packaging film, etc. [72,73]. PS is commonly used in plastic modeling, ionic membranes, disposable cutlery, fishing equipment, etc. [74]. Possible sources of PU include fishing equipment, adhesives, and sealants [75,76]. The present study has highlighted MP contamination in a commercially important fish, *H. nehereus*. It is important to note that the ingestion of MPs by fish can have detrimental effects on their health. MPs can cause physical harm, alter feeding behavior, disrupt digestion, potentially transfer harmful chemicals, and ultimately cause mortality [55].

## 5. Conclusions

The study aimed to assess MP contamination in a commercially important fish, *H. nehereus*. The abundance of MP contamination was found to be 6.98 ± 6.73 MPs/g. Jaffrabad was found to have the highest MP contamination, followed by Jakhau, Mumbai, and Okha. In terms of shape classification, threads were recorded as the dominant shape. Black and blue MPs with a 1–2 mm size were found to be dominant in all the study sites. ATR-FTIR analysis of extorted MPs revealed PE, PS, and PU as polymer compositions of MPs. Based on the identified polymer composition, fishing activities, plastic trash, sealants, and marine equipment can be possible sources of MPs in the ocean. The present study has only assessed MP contamination in commercially important fish. However, more robust information on MP contamination in prey and water may provide a possible pathway for MP transfer. Moreover, the low density of the NaCl solution limits the flotation of high-density plastic polymers. Seasonal variation of MP contamination needs to be assessed to understand the effect of the monsoon on agglomeration and prevalence of MPs. However, the findings of the study highlighted the MP contamination in a commercially important fish, *H. nehereus*, which can be biomagnified in higher taxa and humans as well. Moreover, the study gave insight into the further investigation of the effects of MPs on the physio-behavioral aspects of species. Therefore, efforts to reduce plastic pollution and MP input into the marine environment are crucial to safeguarding marine life and the health of our oceans. The study proposes the implementation of robust collaborative efforts with tourist departments, industries, fishermen, and academic institutions as a means to enhance the efficacy of plastic waste management within the marine ecosystem.

**Supplementary Materials:** The following supporting information can be downloaded at: https://www.mdpi.com/article/10.3390/fishes8090432/s1, Figure S1: Abundance of microplastic contamination in the GT of *H. nehereus* collected from three major fishing harbors of north-west coast of India, showing not a normal distribution (Shapiro-Wilk test, W = 0.73, $p < 0.001$).

**Author Contributions:** Conceptualization, J.T., A.P. and D.K.S.; methodology, K.P. (Krupal Patel); software, D.A.; validation, D.A., S.A. and V.K.Y.; formal analysis, V.R.; investigation, A.P.; resources and data curation, D.K.S.; writing—original draft preparation, K.P. (Kalpana Prusty) and V.R.; writing—review and editing, J.T., A.P. and D.K.S.; visualization, V.K.Y.; supervision and project administration, J.T., A.P. and D.K.S.; funding acquisition, D.A. All authors have read and agreed to the published version of the manuscript.

**Funding:** This research received no external funding.

**Institutional Review Board Statement:** The species does not fall under any schedule of the Indian Wild Life Protection Act, 1972, so ethical approval is not needed for this manuscript.

**Informed Consent Statement:** Not applicable.

**Data Availability Statement:** Data will be made available on request.

**Acknowledgments:** This research was supported by the Researchers Supporting Project number (RSP2023R27), King Saud University, Riyadh, Saudi Arabia. The authors are thankful to the Department of Life Sciences for providing laboratory facilities and infrastructure. The authors would like to thank Heris Patel and Dimple Thacker for their technical support during study periods.

**Conflicts of Interest:** The authors declare no competing financial interest.

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
