# Peer review of "An Assessment of Microplastic Contamination in a Commercially Important Marine Fish, Harpadon nehereus (Hamilton, 1822)"

_fishes, doi:10.3390/fishes8090432_

Round 1

Reviewer 1 Report

This is a highly meaningful and interesting study, as it provides more information to a still scarce research field. However, the study has significant limitations that need to be overcome for clarity and scientific accuracy. The manuscript is well-written, but comprehensive revisions are necessary.

Line 45: Parenthetical Formatting Error

There is a formatting error in the use of parentheses in this line. Please ensure consistent formatting throughout the manuscript.

Consistency in Microplastic Abbreviations

The abbreviation for "microplastics" should remain consistent throughout the manuscript. It appears as "MP" and "MPs" interchangeably. Please standardize its usage.

Line 58: Formatting Error

A formatting error is present in this line. Please revise the formatting to adhere to the required style.

Clarification on Study Materials

Please provide clarity regarding whether the study materials pertain to entire fish or specific tissue groups within fish bodies.

Figure 4(c) Source Clarification

Could you please confirm if Figure 4(c) is a result of your own imaging? Additionally, kindly address why it appears as if pieced together.

Quality Control Measures

It's essential to implement quality control measures for the high-density salt solution used. This includes detecting microplastics or other impurities within the solution, as false positives or negatives may arise.

Filter Paper Quality Control

Similar quality control measures should be applied to the filter paper used. Detecting microplastics or other impurities within the filter paper is crucial to avoid false positive or negative results.

Separation of Microplastics by Density

Considering different types and colors of microplastics, it's recommended to conduct separate density separation using various density salt solutions or oils. This will prevent the omission or confusion of microplastics with varying densities.

Line 263: Spelling Error

There is a spelling error in this line. Please correct it accordingly.

Line 142: Spelling Error

A spelling error is present in this line. Please rectify the error.

Additional References for Microplastic Particle Size

Kindly provide additional references that offer support for the range of microplastic particle sizes mentioned in line 44.

Enhanced Evidence and Description

To strengthen the manuscript, please provide more comprehensive evidence and appropriately detailed descriptions in line 70.

Proper Noun Usage

Upon the first appearance of proprietary names, please spell out their full names rather than using abbreviations, as advised in line 114.

Microplastic Size and Food Ingestion Relation

Could you elaborate on the potential relationship between microplastic size and ingested food, as well as the size of fish?

Enhanced Significance Visibility in Figure 2

To enhance clarity, consider directly annotating levels of significance (a, b, c) onto the bar chart in Figure 2 for better interpretation.

Suggestion for Additional Significance Analysis in Figure 6

Considering your suggestion, consider adding a significance analysis graph for different particle size ranges in Figure 6.

Line 178: Full Noun Spelling

Upon initial introduction, please spell out the full name of the noun mentioned in line 178.

Lines 231-235: Repetition with Results

The content in these lines appears to repeat the description provided in the results section. Please rephrase or remove redundant information.

Lines 199-228: Depth of Discussion

Please consider expanding the discussion in lines 199-228 to move beyond result description and into substantive analysis.

Line 256: Increased Supporting Evidence

To substantiate the claim, more evidence is needed for the statement in line 256.

Line 262: Irrelevant Content

The content in line 262 appears to be disconnected from the surrounding context. Please revise accordingly.

Conclusion Section Enhancement

The conclusion section requires a concise summary of the main objectives, methods, results, and contributions. Avoid repeating content from the results section. Additionally, discuss limitations, future research directions, and policy recommendations for a more comprehensive conclusion.

Line 303: Incomplete Citation

Please review and ensure that the citation in line 303 is complete.

Line 308: Incomplete Citation

Please verify the citation in line 308 for completeness.

Line 329: Italics for Latin Names

Ensure that Latin names are in italics throughout the text. Please review line 329 and the entire manuscript for proper formatting.

Line 379: Consistent Capitalization

Maintain consistent capitalization throughout the manuscript, including line 379. Please review for uniformity.

Lines 447 and 448: Incomplete Citations

Please verify and complete the citations in lines 447 and 448.

Requires major revisions

Author Response

Response to Reviewer

Comments and Suggestions for Authors

This is a highly meaningful and interesting study, as it provides more information to a still scarce research field. However, the study has significant limitations that need to be overcome for clarity and scientific accuracy. The manuscript is well-written, but comprehensive revisions are necessary.

We would like to thank the esteemed reviewer for his insightful remarks and excellent ideas, which have really aided us in improving our manuscript. The changes that were made are listed below and the revised sentences of manuscript are highlighted with yellow colour:

Comment 1: Line 45: Parenthetical Formatting Error

There is a formatting error in the use of parentheses in this line. Please ensure consistent formatting throughout the manuscript.

Author Response: The authors have revised as (>1 μm ~ <5 mm) as suggested by respected reviewer.

Comment 2: Consistency in Microplastic Abbreviations

The abbreviation for "microplastics" should remain consistent throughout the manuscript. It appears as "MP" and "MPs" interchangeably. Please standardize its usage.

Author Response: Abbreviation of microplastics as MPs incorporated throughout manuscript as suggested by respected reviewer.

Comment 3: Line 58: Formatting Error

A formatting error is present in this line. Please revise the formatting to adhere to the required style. 

Author Response: MP’ is replaced with MPs in revised version of manuscript as suggested by the respected reviewer.

Comment 4: Clarification on Study Materials

Please provide clarity regarding whether the study materials pertain to entire fish or specific tissue groups within fish bodies.

Author Response: As per suggestion of respected reviewer, Assessment of MPs have been investigated in gastrointestinal tract (GT) of fish. The sentence added as follow,

“Each fish was dissected in a metal tray and GT was isolated to check the contamination of MPs.”

Comment 5: Figure 4(c) Source Clarification

Could you please confirm if Figure 4(c) is a result of your own imaging? Additionally, kindly address why it appears as if pieced together.

Author Response: The authors would like to thanks the reviewer. To avoid misidentification of MPs, the following criteria were considered for the identification of MPs: no cellular structure, MPs threads should be equally thick throughout the length. Moreover, hot needing technique was used for identification of film, foam and fragment. Two pieced of films were placed together for photography of extracted MPs.

Comment 6: Quality Control Measures

It's essential to implement quality control measures for the high-density salt solution used. This includes detecting microplastics or other impurities within the solution, as false positives or negatives may arise.

Author Response: To avoid the laboratory contamination, laboratory blanks having salt solution were run with each slot. No MPs were found in blanks.

Comment 7: Filter Paper Quality Control

Similar quality control measures should be applied to the filter paper used. Detecting microplastics or other impurities within the filter paper is crucial to avoid false positive or negative results.

Author Response: To avoid the laboratory contamination, laboratory blanks having salt solution were run with each slot. The blank filter papers had no MPs contamination. The above sentence is mentioned in contamination control section.

Comment 8: Separation of Microplastics by Density

Considering different types and colors of microplastics, it's recommended to conduct separate density separation using various density salt solutions or oils. This will prevent the omission or confusion of microplastics with varying densities.

Author Response: Floatation of MPs with supersaturated NaCl was carried out as previously used protocol by Robin et al., 2019, Rabari et al., 2022 and 2023.

Comment 9: Line 263: Spelling Error

There is a spelling error in this line. Please correct it accordingly.

Author Response: Corrected as per comment of the respected reviewer.

Comment 10: Line 142: Spelling Error

A spelling error is present in this line. Please rectify the error.

Author Response: Corrected as per comment of the respected reviewer.

Comment 11: Additional References for Microplastic Particle Size

Kindly provide additional references that offer support for the range of microplastic particle sizes mentioned in line 44.

Author Response: Additional references are added as per comment of the respected reviewer.

Comment 12: Enhanced Evidence and Description

To strengthen the manuscript, please provide more comprehensive evidence and appropriately detailed descriptions in line 70.

Author Response: Revised as per comment of the respected reviewer.

Comment 13: Proper Noun Usage

Upon the first appearance of proprietary names, please spell out their full names rather than using abbreviations, as advised in line 114.

Author Response: Abbreviation of gastrointestinal tract as GT was used during its first appearance in introduction section.

Comment 14: Microplastic Size and Food Ingestion Relation

Could you elaborate on the potential relationship between microplastic size and ingested food, as well as the size of fish?

Author Response: The regression analysis between fish length and MPs abundance have been incorporated as per comment of the learned reviewer.

Comment 15: Enhanced Significance Visibility in Figure 2

To enhance clarity, consider directly annotating levels of significance (a, b, c) onto the bar chart in Figure 2 for better interpretation.

Author Response: Level of significance: ** p < 0.01 is added in figure and in its caption as suggested by respected reviewer.

Comment 16: Suggestion for Additional Significance Analysis in Figure 6

Considering your suggestion, consider adding a significance analysis graph for different particle size ranges in Figure 6.

Author Response: The significant difference between size classes of MPs calculated as per comment of the respected reviewer.

Comment 17: Line 178: Full Noun Spelling

Upon initial introduction, please spell out the full name of the noun mentioned in line 178.

Author Response: Revised as per comment of the respected reviewer.

Comment 18: Lines 231-235: Repetition with Results

The content in these lines appears to repeat the description provided in the results section. Please rephrase or remove redundant information.

Author Response: The content reframed as follow,

The study site Jaffrabad has shown higher abundance of MPs contamination followed by, Jakhau and Okha. Moreover, significant contamination was recorded between the study sites.

Comment 19: Lines 199-228: Depth of Discussion

Please consider expanding the discussion in lines 199-228 to move beyond result description and into substantive analysis.

Author Response: The discussion part revised as suggested by the respected reviewer.

Comment 20: Line 256: Increased Supporting Evidence

To substantiate the claim, more evidence is needed for the statement in line 256.

Author Response: Thank you for the comments. The sentence has been removed from manuscript due to lack of strong scientific evidence.

Comment 21: Line 262: Irrelevant Content

The content in line 262 appears to be disconnected from the surrounding context. Please revise accordingly.

Author Response: The authors have revised the content as follow,

Moreover, it was found that MPs can act as carriers for other harmful pollutants, such as toxic chemicals (polychlorinated biphenyls, plasticizers and poly-brominated diphenyl ether) and heavy metals, which can attach to the surface of the particles.

Comment 22: Conclusion Section Enhancement

The conclusion section requires a concise summary of the main objectives, methods, results, and contributions. Avoid repeating content from the results section. Additionally, discuss limitations, future research directions, and policy recommendations for a more comprehensive conclusion.

Author Response: The conclusion is reframed as well as limitations, future directions and policy recommendation added as per suggestion of respected reviewer.

Comment 23: Line 303: Incomplete Citation

Please review and ensure that the citation in line 303 is complete.

Author Response: Formatting style of each reference checked twice. Moreover, DOI numbers were added using Mendeley as suggested by respected reviewer.

Comment 24: Line 308: Incomplete Citation

Please verify the citation in line 308 for completeness.

Author Response: Formatting style of each reference checked twice. Moreover, DOI numbers were added using Mendeley as suggested by respected reviewer.

Comment 25: Line 329: Italics for Latin Names

Ensure that Latin names are in italics throughout the text. Please review line 329 and the entire manuscript for proper formatting.

Author Response: The authors have revised as per comment of the respected reviewer.

Comment 26: Line 379: Consistent Capitalization

Maintain consistent capitalization throughout the manuscript, including line 379. Please review for uniformity.

Author Response: The authors have revised as per comment of the respected reviewer.

Comment 27: Lines 447 and 448: Incomplete Citations 

Please verify and complete the citations in lines 447 and 448.

Author Response: Formatting style of each reference checked twice. Moreover, DOI numbers were added using Mendeley as suggested by respected reviewer.

Reviewer 2 Report

Comments on the manuscript,

An assessment of microplastic contamination in commercially 2 important marine fish, Harpadon nehereus (Hamilton, 1822)., by Prusty et al.

Recently, many articles have been calling attention to possible effects of microplastics in fish and marine organisms. In this work, MP 15 contamination was investigated in Harpadon nehereus collected from principle fishing harbors of the 16 north-west coast, India. The topic is good and the manuscript were written well to confirm all collected specimens of Harpadon nehereus were polluted by MPs in India. The reuslts provide the fact that a greater menace to sea food safety due to trophic transfer, which may cause hazardous effect to human health. Howver, the presentation should be revised and some portions need to be improved. I recommend publication after a moderate revision. Belows are specific comments:

Abstract, line 28, “tropical transfer” should read “trophic transfer”.

Introduction,

Line 45, MPs (>1 μm–<5), mm) should be (>1 μm ~ <5 mm)  

line 56 to 70, The authors described MP has been recorded in fish too. I think the authors should include “the behavior of Harpadon nehereus and breifly describe they like to eat little fish and crustaceous etc in the Introduction. For example, recent reseach reavels that some little fish containing microplastics are widely found in many Asian countries (Piyawardhana et al., 2022). The results can support the findings that these MP can enter marine food chain throught trophic transfer. I suggest that the authors should include some recent report about commercial fish products investigated in several Asian countries and dsicuss potential MP transfer from little fish to Harpadon nehereus

Piyawardhana et al., (2022) Occurrence of microplastics in commercial marine dried fish in Asian countries. Journal of Hazardous Materials. 423, 127093.

The authors should include few in depth discussions regarding factors influencing MP transportation in basic marine food web such as from zooplankton to shrimp or little fish because these shrimp and/or carnivorous and omnivorous fish are likely to eat phytoplantkon and zooplankton, and they need to be included in the revised verions. Otherwise, the mansucript looks like a technique report.

 Table 2’s format need to be revised as a whole vocubulary. Also, MP in fishes (MPs/g) exists in subtitle, it does not need to show (MPs/g) after MP values. ONLY “6.98 ± 6.73”. is fine, same as other numbers.

Author Response

Response to reviewer

Recently, many articles have been calling attention to possible effects of microplastics in fish and marine organisms. In this work, MP 15 contamination was investigated in Harpadon nehereus collected from principle fishing harbors of the 16 north-west coast, India. The topic is good and the manuscript were written well to confirm all collected specimens of Harpadon nehereus were polluted by MPs in India. The reuslts provide the fact that a greater menace to sea food safety due to trophic transfer, which may cause hazardous effect to human health. Howver, the presentation should be revised and some portions need to be improved. I recommend publication after a moderate revision. Belows are specific comments:

We would like to thank the esteemed reviewer for his insightful remarks and excellent ideas, which have really aided us in improving our manuscript. The changes that were made are listed below and the revised sentences of manuscript are highlighted with bright green colour:

Comment 1: Abstract, line 28, “tropical transfer” should read “trophic transfer”.

Author Response: “tropical transfer” is replaced by “trophic transfer” as per comment of the respected reviewer.

Introduction,

Comment 2: Line 45, MPs (>1 μm–<5), mm) should be (>1 μm ~ <5 mm).

Author Response: Revised as (>1 μm ~ <5 mm) as per the comments of respected reviewer.

Comment 3: Line 56 to 70, The author’s described MP has been recorded in fish too. I think the authors should include “the behavior of Harpadon nehereus and breifly describe they like to eat little fish and crustaceous etc in the Introduction. For example, recent reseach reavels that some little fish containing microplastics are widely found in many Asian countries (Piyawardhana et al., 2022). The results can support the findings that these MP can enter marine food chain throught trophic transfer. I suggest that the authors should include some recent report about commercial fish products investigated in several Asian countries and dsicuss potential MP transfer from little fish to Harpadon nehereus

Piyawardhana et al., (2022) Occurrence of microplastics in commercial marine dried fish in Asian countries. Journal of Hazardous Materials. 423, 127093.

Author Response: Revised as follow,

Small fishes and prawns were recorded dominantly in gut contain of H. nehereus (Ghosh, 2014). Recently, studies have revealed the MPs contamination in small fishes and prawns of India (Piyawardhana et al., 2022; Daniel et al., 2020 54). MPs ingestion by small fishes and prawns can be transferred to the H. nehereus via food chain. Resulting, it was quite imperative to assess the MPs contamination in gastrointestinal tract (GT) of H. nehereus.

Comment 4: The authors should include few in depth discussions regarding factors influencing MP transportation in basic marine food web such as from zooplankton to shrimp or little fish because these shrimp and/or carnivorous and omnivorous fish are likely to eat phytoplantkon and zooplankton, and they need to be included in the revised verions. Otherwise, the mansucript looks like a technique report.

Author Response: The discussion part revised as per given comment of the respected reviewer.

Comment 5: Table 2’s format need to be revised as a whole vocubulary. Also, MP in fishes (MPs/g) exists in subtitle, it does not need to show (MPs/g) after MP values. ONLY “6.98 ± 6.73”. is fine, same as other numbers.

Author Response: Unit MPs/g is removed from table as per comment of the respected reviewer.

Reviewer 3 Report

This study, "An assessment of microplastic contamination in commercially important marine fish, Harpadon nehereus (Hamilton, 1822)", investigated microplastic ingestion by marine fish in places along the Indian coastline. This study contributes to our knowledge of microplastic contamination in seafood. However, the manuscript needs some improvement before being considered for publication. Comments included below: 

1.   MPs are a threat to fish in aquatic environments not only marine ones. "global threat to marine organism fish" should be replaced with " to fish".

2.   Better delete reference from the abstract. Replace with a sentence such as "using previously documented protocol". 

3.   The sentence " the findings highlighted a greater menace to seafood safety due to tropical transfer, which cause hazardous effect to human health" does not have clear evidence in the manuscript. How did you prove the tropical transfer and hazardous effects on human health? 

4.   Limitations for using NaCl solution for flotation of MPs should be discussed as high-density MPs might not be recovered. 

5.    How did the authors calculate MP per gram? How many MPs were calculated per fish? 

6.   Did the authors find a relationship between fish size and MPs detected? 

Acceptable 

Author Response

Response to reviewer

Comments and Suggestions for Authors

This study, "An assessment of microplastic contamination in commercially important marine fish, Harpadon nehereus (Hamilton, 1822)", investigated microplastic ingestion by marine fish in places along the Indian coastline. This study contributes to our knowledge of microplastic contamination in seafood. However, the manuscript needs some improvement before being considered for publication. Comments included below:

We would like to thank the esteemed reviewer for his insightful remarks and excellent ideas, which have really aided us in improving our manuscript. The changes that were made are listed below and the revised sentences of manuscript are highlighted with turquoise colour

Comment 1:  MPs are a threat to fish in aquatic environments not only marine ones. "global threat to marine organism fish" should be replaced with " to fish".

Author Response: “to fish” added in place of “to marine organism fish” as per comment of the respected reviewer.

Comment 2:   Better delete reference from the abstract. Replace with a sentence such as "using previously documented protocol".

Author Response: Citations deleted from abstract as per comment of the respected reviewer.

Comment 3:   The sentence "the findings highlighted a greater menace to seafood safety due to tropical transfer, which cause hazardous effect to human health" does not have clear evidence in the manuscript. How did you prove the tropical transfer and hazardous effects on human health?

Author Response: Sentences are reframed as follow,

Consequently, accumulated MPs can be transfer to the higher taxa due to bioaccumulation. Moreover, when humans consume fish contaminated with MPs, there is a potential for the transfer of these particles from the fish's tissues to human bodies.

Comment 4:   Limitations for using NaCl solution for flotation of MPs should be discussed as high-density MPs might not be recovered.

Author Response: Floatation of MPs with supersaturated NaCl was carried out as previously used protocol by Robin et al., 2019, Rabari et al., 2022 and 2023.

Comment 5:   How did the authors calculate MP per gram? How many MPs were calculated per fish?

Author Response: Total count of MPs divided by wet weight of gut (g) was considered as MPs/g. Total count of MPs recorded in a gut of fish was considered as MPs/individual. The above calculated is followed to previously documented articles.

Comment 6:   Did the authors find a relationship between fish size and MPs detected?

Author Response: The regression analysis between fish length and MPs abundance have been incorporated as per comment of the respected reviewer.

Round 2

Reviewer 1 Report

Accepted

Minor editing of English required

Author Response

We would like to thank the esteemed reviewer for his insightful remarks and excellent ideas, which have really aided us in improving our manuscript. We have gone through the comments and revised accordingly. The changes that were made are listed below and the revised sentences of manuscript are highlighted with turquoise colour:

Minor editing of English required

Author Response: Thank you for your valuable comment. English editing has been done in revised manuscript thoroughly as suggested by the respected reviewer.

Reviewer 3 Report

The authors responded to the comments, but this article still needs some clarifications. For instance, line 126 why supersaturated, not saturated solution. What do you mean by supersaturated? Do you mean  (1.2 g/cm3)? Furthermore, despite many studies using saturated NaCl solution, some polymers (principally polyester, PVC, and PET) have a higher density than NaCl and, therefore, can not be separated using this approach. This limitation should be included/discussed. 

Line 164: Blank replicates are used to detect contamination, not avoid contamination. Please correct. 

Line 174: correct correction to correlation 

Acceptable

Author Response

Respected Reviewer,

We would like to thank the esteemed reviewer for his insightful remarks and excellent ideas, which have really aided us in improving our manuscript. We have gone through the comments and revised accordingly. The changes that were made are listed below and the revised sentences of manuscript are highlighted with yellow colour:

Comment 1: The authors responded to the comments, but this article still needs some clarifications. For instance, line 126 why supersaturated, not saturated solution. What do you mean by supersaturated? Do you mean  (1.2 g/cm3)? Furthermore, despite many studies using saturated NaCl solution, some polymers (principally polyester, PVC, and PET) have a higher density than NaCl and, therefore, can not be separated using this approach. This limitation should be included/discussed. 

Author Response: A supersaturated solution contains more dissolved solute than is required for preparing a saturated solution until the solute remains undissolved in solution. The limitation of the NaCl solution is added to the manuscript as per the comment of a respected reviewer. Kindly check line 321–322.

Comment 2: Line 164: Blank replicates are used to detect contamination, not avoid contamination. Please correct. 

Author Response: The sentence has been revised as per the comment of a respected reviewer. Kindly check line 145–147.

Comment 3: Line 174: correct correction to correlation.

Author Response: “Correction” is replaced by “correlation”; thank you for your kind input. Kindly check line 175.